# Thrust Vectoring Control of a Novel Tilt-Rotor UAV Based on Backstepping Sliding Model Method

**DOI:** 10.3390/s23020574

**Published:** 2023-01-04

**Authors:** Zelong Yu, Jingjuan Zhang, Xueyun Wang

**Affiliations:** 1School of Instrumentation Science and Opto-Electronics Engineering, Beihang University, Beijing 100191, China; 2School of Aeronautic Science and Engineering, Beihang University, Beijing 100191, China

**Keywords:** backstepping method, controller design, sliding mode control, thrust vectoring control, tilt-rotor

## Abstract

In this paper, a control method of a novel tilt-rotor UAV with a blended wing body layout is studied. The novel UAV is capable of vertical take-off and landing and has strong stealth capabilities that can be applied to carrier-borne reconnaissance aircraft. However, the high aspect ratio of blended wing body UAVs leads to a wingtip or oar-tip touchdown problem when adopting the conventional position-attitude control (CPAC) scheme with a large crosswind disturbance. Moreover, when the UAV is subject to interference during reconnaissance, aerial photography, and landing missions, the conventional scheme cannot provide both attitude stability and track accuracy. First, a direct thrust vectoring control (DTVC) scheme is proposed. The control authority of the rotor tilt mechanism was added to enable the decoupling of the attitude and trajectory and to improve the response rate and response bandwidth of the flight trajectory. Second, considering the problems of strong couplings and parameter uncertainties and the nonlinear features and mismatched perturbations that are inevitable in the tilt-rotor, we designed a robust UAV controller based on the backstepping sliding mode control method and determined the stability of the control system through the Lyapunov function. Finally, in the case of crosswire interference during vertical takeoff and landing and the aerial photography missions of the UAV, the numerical simulation of the CPAC scheme and the DTVC scheme was carried out, respectively, and the Monte Carlo random test method was introduced to conduct the statistical test of the landing accuracy. The simulation results show that the DTVC scheme improves the landing accuracy and speed compared to the CAPC scheme and decouples the position control loop from the attitude control loop, finally enabling the UAV to complete the flight control in the VTOL phase.

## 1. Introduction

In recent years, shipboard UAVs have attracted extensive attention. With the continuous development of UAVs’ technology and the expansion of their mission execution capability, ocean-going operations of the UAVs have become usual [1]. As a novel structural aircraft, the tilt-rotor UAV has become a focus of shipboard aircraft research [2]. Compared with helicopters, the tilt-rotor UAVs have the characteristics of a wider flight envelope and greater load capacity. In contrast to fixed-wing aircraft, tilt-rotor UAVs can perform take-off and landing (VTOL) without a runway [3]. Therefore, tilt-rotor UAVs are more suitable for shipboard applications than conventional aircraft.

With the development of a UAV design and control technology, some novel tilt-rotor UAV structures have been developed in recent years [4]. In [5,6], a quadcopter, allowing its rotors to tilt around the main axes, was designed. The UAV improved mobility and trajectory tracking capabilities. A quad tilt-wing UAV was designed and studied in [7,8]. Its rotors and wings can tilt together to reduce the impact caused by the wing occlusion effect. In [9,10], a tri-tilt-rotor fixed-wing configuration was adopted to design the UAV, consisting of two main tilting rotors at the front sides of the fuselage and a tail tilting rotor. This design can reduce aerodynamic interference by separating the rotors and wings.

After evaluating the advantages and disadvantages of the different types of VTOL UAVs, a novel tilt-rotor UAV is presented in this paper. The prototype is shown in Figure 1. Similar to the tilt tri-rotor, it has three rotors with tilt mechanisms and two elevons for all flight modes. The difference is that the novel configuration adopts the blended wing body layout and only uses part of the wing tilts with the rotor. This design minimizes aerodynamic drag and increases endurance, on top of improving the UAV’s stealth performance [11].

Currently, the conventional control scheme of the underactuated system has been applied to the control system design of most tilt-rotor aircraft, in which the attitude inner loop controls the position outer loop, and the vertical altitude control loop is added [12,13]. In dynamic modeling, the common rotorcraft modeling approach is used, where the attitude is assumed to be stable and constant, and the effect of changing the attitude on the UAV’s stability and safety during take-off and landing is ignored. There is no more accurate control scheme based on the multi-actuator properties of tilt-rotor UAVs [14,15].

However, the oceanic operation of an aircraft is subject to a more severe environment than its land operation. The control difficulties and risks are extreme for the shipboard UAV in the VTOL phase, as the area of the shipboard is limited, the wind is not uniform, and the shipboard motion occurs in roll, pitch, and heave [16,17]. At the same time, the UAV with a blended wing body layout has a longer wing. When crosswinds are present, and a conventional control scheme is adapted for landing, a roll angle is required to balance the aerodynamic forces due to the sideslip. This approach obviously increases the risk of the wing tips hitting the shipboard surface [18]. Second, the aerodynamic interference is complicated at the beginning of the tilt-rotor’s tilting transition process, and the control strategy is the same as in the VTOL phase [19]. However, when the conventional control strategy is used for position control, the attitude of the UAV needs to be changed, which leads to a more complex interference of the UAV and increases the control difficulty. Therefore, keeping the attitude of the UAV stable during the initial stages of its tilting transition process will considerably increase its safety [20]. Third, the UAV needs to maintain the stability of the camera platform when carrying out aerial shooting missions along the desired trajectory. When the trajectory changes due to wind field interference, the conventional control scheme changes the attitude of the UAV for trajectory correction. Therefore, the camera platform needs to adjust the angle to maintain the stability of aerial photography [21].

To sum up, the conventional control scheme for this UAV has its own shortcomings.

A novel thrust vectoring control scheme of tilt-rotor UAVs is proposed to solve the above problems [22,23]. This scheme increases the control authority of the two tilt mechanisms of the main rotor, designs a reasonable control allocation strategy, decouples the attitude control loop from the position control loop while maintaining the UAV in a stable attitude, and directly controls the thrust vector for trajectory tracking. The UAV can have a stable attitude for aerial photography, a safe tilt transition process, and an accurate and stable landing.

Additionally, tilt-rotor UAVs have more complex characteristics than regular aircraft, which makes them profoundly nonlinear and coupled systems with model uncertainties and a complex aerodynamic interaction, which increases the complexity and difficulty of the control system design. It is well understood that the PID control method is still widely used in engineering practice. The PID controller of roll, pitch, and yaw attitude loops was designed in [24], and the effectiveness of the controller was verified by flight experiments. There are similar studies in the same type of literature [25]. However, the PID control method cannot deal with the problem of actuator saturation and uncertain perturbations entirely.

Thus, designing a better controller to improve the control performance of tilt-rotor UAVs motivates this paper. Recently, various nonlinear control methods have been studied in the tilt-rotor UAV controller design, such as sliding mode control (SMC) [26], adaptive control (AC) [27], model predictive control (MPC) [22], and linear quadratic regulator control (LQRC) [28]. SMC has been widely used in control system designs for various aircraft due to its strong robustness and resistance to disturbance. However, the chattering phenomenon of the system is the main drawback of SMC [29], which can be solved by designing an appropriate approach law method [30]. The relatively long asymptotic convergence property of SMC is another weakness besides the chattering phenomenon. The backstepping recursive control structure is designed to avoid this problem on the premise of ensuring a finite-time convergence while maintaining stability [31].

In view of that, an improved flight control law based on the combination of sliding mode control with a backstepping recursive control structure is proposed to design a controller for the novel tilt-rotor UAV. In this paper, the thrust vectoring control of the novel tilt-rotor UAV based on the backstepping sliding model method is researched. The main contributions of the paper include:Firstly, aiming at the requirements of strong stealth and vertical take-off and landing characteristics of carrier-based UAVs, a novel tilt-rotor aircraft with a blende wing body layout is introduced, and its dynamics are modeled;Second, a novel thrust vectoring control scheme of the novel tilt-rotor UAV is proposed to solve the problems of the wing tip touch and coupling of the attitude and position in the conventional control. This scheme increases the control authority of the two tilt mechanisms of the main rotor, designs a reasonable control allocation strategy, and decouples the attitude control loop from the position control loop;Finally, a flight control law based on the combination of the sliding-mode control with a backstepping recursive control structure is proposed to design a controller for the novel tilt-rotor UAV. Moreover, based on mathematical modeling, the proposed control law is used to simulate and compare the control effects of the two control schemes in the case of a target point flight, flight with attitude disturbance, and landing with crosswind disturbance. The effectiveness of the proposed control algorithm and scheme is demonstrated.

The rest of the paper is organized as follows. In Section 2, the nonlinear dynamic model of the novel tilt-rotor UAV is introduced. A controller design method based on the combination of SMC with a backstepping recursive control structure is discussed, and a thrust vectoring control scheme is proposed in Section 3. In Section 4, the comparative simulation demonstrations are given. Finally, the conclusions are drawn in Section 5.

## 2. Mathematical Modeling

### 2.1. Description of Model

The work presented in this paper focuses on a novel tilt-rotor UAV, depicted in Figure 1. It has three rotors and two elevons with their own tilting mechanisms to achieve the vertical take-off and landing (VTOL) phase, tilting transition (TT) phase, and fixed-wing (FW) phase. The two main rotors fixed to the forward part of the vehicle frame rotate in opposite directions, reducing the resulting reaction torque to approximately zero.

Moreover, the novel UAV is equipped with three rotor tilt mechanisms and is, therefore, capable of a flying mode transition from the VTOL phase to the FW phase. Unlike common UAVs, this aircraft has a flying wing configuration, which increases the lift-to-drag ratio. Additionally, the normal horizontal wake, elevator, and rudder are eliminated, which means fewer mechanisms, a higher energy efficiency ratio, and stealthy performance.

### 2.2. Definition of Coordinate System

Figure 2 shows the structure of the novel tilt-rotor UAV. The definitions of the main wing body reference frames and the corresponding conversion matrices can be defined as follows: $O_{b}=X_{b}Y_{b}Z_{b}$ is the body-fixed coordinates frame (BFF), $O_{b}$ is at the center of gravity, $X_{b}$ is pointing to the right of the UAV, $Y_{b}$ is pointing front, and $Z_{b}$ is pointing down. $O_{mr}=X_{mr}Y_{mr}Z_{mr}$, $O_{ml}=X_{ml}Y_{ml}Z_{ml}$, and $O_{mb}=X_{mb}Y_{mb}Z_{mb}$ are the motor coordinate frames, which are fixed with three rotors, respectively. $O_{n}=X_{n}Y_{n}Z_{n}$ is the navigation coordinates frame, which coincides with the North–East–Down geographic coordinates frame (NED). $O_{b}$ is the original point of the tilt-rotor UAV, and the coordinate is ${\left(0,0,0\right)}^{T}$. Therefore, the coordinate of the right rotor is ${\left(x_{r},y_{r},z_{r}\right)}^{T}$, the left one is ${\left(x_{l},y_{l},z_{l}\right)}^{T}$, and the rear one is ${\left(x_{b},y_{b},z_{b}\right)}^{T}$.

The transformation matrices among the coordination frames can be described by
(1)$$\begin{array}{l} \left\{\begin{array}{l} C_{n}^{b}=\left[\begin{array}{ccc} c\theta c\psi & c\theta s\psi & -s\theta \\ s\theta c\psi s\phi -s\psi c\phi & s\theta s\psi s\phi +c\psi c\phi & c\theta s\phi \\ s\theta c\psi c\phi +s\psi s\phi & s\theta s\psi c\phi -c\psi s\phi & c\theta c\phi \end{array}\right]\\ C_{b}^{n}=\left[\begin{array}{lll} c\theta c\psi & s\theta c\psi s\phi -s\psi c\phi & s\theta c\psi c\phi +s\psi s\phi \\ c\theta s\psi & s\theta s\psi s\phi +c\psi c\phi & s\theta s\psi c\phi -c\psi s\phi \\ -s\theta & \;\;c\theta s\phi & \;\;c\theta c\phi \end{array}\right] \end{array}\right.\\ \left\{\begin{array}{l} C_{rm}^{b}=\left[\begin{array}{ccc} cA_{r} & 0 & -sA_{r}\\ 0 & 1 & 0\\ sA_{r} & 0 & cA_{r} \end{array}\right]\\ C_{lm}^{b}=\left[\begin{array}{ccc} cA_{l} & 0 & -sA_{l}\\ 0 & 1 & 0\\ sA_{l} & 0 & cA_{l} \end{array}\right]\\ C_{bm}^{b}=\left[\begin{array}{ccc} 1 & 0 & 0\\ 0 & cA_{b} & -sA_{b}\\ 0 & sA_{b} & cA_{b} \end{array}\right] \end{array}\right. \end{array}$$
where ϕ,θ, and ψ represent the Euler angles, $A_{r}$,$A_{l}$, and $A_{b}$ are the tilting angles of the rotors, c and s are the symbols of the cosine function and the sine function, respectively, and $C_{mr}^{b}$, $C_{ml}^{b}$, and $C_{mb}^{b}$ are the transformation matrices among the motor coordinate frames and the BFF.

### 2.3. Dynamic Model

Utilizing the standard Newton–Euler formulation, the equation of the dynamics for the UAV is described as follows:(2)$$\begin{array}{l} \left\{\begin{array}{l} {\vec{F}}^{b}=\frac{d}{dt}(m{\vec{V}}^{b})=m\left(\frac{\delta {\vec{V}}^{b}}{\delta t}+{\vec{\Omega }}^{b}\times {\vec{V}}^{b}\right)\\ {\vec{M}}^{b}=\vec{I}\left(\frac{\delta {\vec{\Omega }}^{b}}{\delta t}\right)+{\vec{\Omega }}^{b}\times \left(\vec{I}{\vec{\Omega }}^{b}\right) \end{array}\right.\\ \left\{\begin{array}{l} {\vec{\dot{X}}}^{n}=C_{n}^{b}{\vec{V}}^{b}\\ \vec{\dot{\Theta }}=J_{n}^{b}{\vec{\Omega }}^{b} \end{array}\right. \end{array}$$
where ${\vec{F}}^{b}$ is the BFF-based total force vector, ${\vec{M}}^{b}$ is the BFF-based total moment vector, $\vec{V}={\left[\begin{array}{ccc} u & v & w \end{array}\right]}^{T}$ is the BFF-based velocity vector, $\;{\vec{\Omega }}^{b}={\left[\begin{array}{ccc} p & q & r \end{array}\right]}^{T}$ is the BFF-based angular rotation rate vector, ${\vec{X}}^{n}={\left[\begin{array}{ccc} x^{n} & y^{n} & z^{n} \end{array}\right]}^{T}$ is the NED-based position vector, and $\vec{\Theta }={\left[\begin{array}{ccc} \phi & \theta & \psi \end{array}\right]}^{T}\;$ is the Tait–Bryan rotational angle vector. Additionally, m is the total mass, and $\vec{I}$ is the moment of the inertia matrix. Furthermore, $C_{n}^{b}$ is the BFF-NED translational velocities transformation matrix, and $J_{n}^{b}$ is the Tait–Bryan rotational angle rate transformation matrix.

In the VTOL phase, the aerodynamic forces of the tilt-rotor UAV are reduced or null due to the low airspeed, and the position and attitude of the UAV are mainly controlled by the rotors and tilting mechanisms. Therefore, a simplistic assumption, taking no account of the aerodynamic forces, may be taken, as will be the case here.

This paper focuses on the advantage of thrust vectoring control over conventional control for the novel tilt-rotor UAV. The simulations can be performed directly with the thrust of the three rotors as the variables without affecting the conclusions. Therefore, the rotor thrust of the UAV can be modeled assuming that both the propeller and ducted fan can provide sufficient thrust without the necessity of modeling it separately.

To sum up, the resultant force vector, ${\vec{F}}^{b}={\left[\begin{array}{ccc} {\vec{F}}_{x}^{b} & {\vec{F}}_{y}^{b} & {\vec{F}}_{z}^{b} \end{array}\right]}^{T}$, in the BBF is produced from the rotor thrusts and gravitational force, and it can be described as
(3)$$\begin{array}{l} \left\{\begin{array}{l} {\vec{T}}_{r}^{b}=C_{rm}^{b}\cdot {\left[\begin{array}{ccc} 0 & 0 & T_{rm} \end{array}\right]}^{T}\\ {\vec{T}}_{l}^{b}=C_{lm}^{b}\cdot {\left[\begin{array}{ccc} 0 & 0 & T_{lm} \end{array}\right]}^{T}\\ {\vec{T}}_{b}^{b}=C_{bm}^{b}\cdot {\left[\begin{array}{ccc} 0 & 0 & T_{bm} \end{array}\right]}^{T}\\ {\vec{G}}^{b}=C_{n}^{b}\cdot {\left[\begin{array}{ccc} 0 & 0 & mg \end{array}\right]}^{T} \end{array}\right.\\ {\vec{F}}^{b}={\vec{T}}_{r}^{b}+{\vec{T}}_{l}^{b}+{\vec{T}}_{b}^{b}+{\vec{G}}^{b} \end{array}$$
where ${\vec{T}}_{r}^{b}$,${\vec{T}}_{l}^{b}$, and ${\vec{T}}_{b}^{b}$ are the thrust vectors generated by the rotors in the BFF coordinate frame, ${\vec{G}}^{b}$ represents the gravitational vectors in the BFF coordinate frame, and $T_{rm}$, $T_{lm}$, and $T_{bm}$ are scalars, indicating the thrusts of the propeller.

The total moment vector, ${\vec{M}}^{b}={\left[\begin{array}{ccc} {\vec{M}}_{x}^{b} & {\vec{M}}_{y}^{b} & {\vec{M}}_{z}^{b} \end{array}\right]}^{T}$, generated by the propulsion systems, can be expressed as Equation (4):(4)$$\begin{array}{l} \left\{\begin{array}{l} {\vec{M}}_{r}^{b}={\left[\begin{array}{ccc} x_{r} & y_{r} & z_{r} \end{array}\right]}^{T}\times T_{r}^{b}\\ {\vec{M}}_{l}^{b}={\left[\begin{array}{ccc} x_{l} & y_{l} & z_{l} \end{array}\right]}^{T}\times T_{l}^{b}\\ {\vec{M}}_{b}^{b}={\left[\begin{array}{ccc} x_{b} & y_{b} & z_{b} \end{array}\right]}^{T}\times T_{b}^{b}\\ {\vec{M}}_{rt}^{b}=C_{rm}^{b}\cdot {\left[\begin{array}{lll} 0 & 0 & -Q_{r} \end{array}\right]}^{T}\\ {\vec{M}}_{lt}^{b}=C_{lm}^{b}\cdot {\left[\begin{array}{lll} 0 & 0 & Q_{l} \end{array}\right]}^{T}\\ {\vec{M}}_{bt}^{b}=C_{bm}^{b}\cdot {\left[\begin{array}{lll} 0 & 0 & -Q_{b} \end{array}\right]}^{T} \end{array}\right.\\ {\vec{M}}^{b}={\vec{M}}_{r}^{b}+{\vec{M}}_{l}^{b}+{\vec{M}}_{b}^{b}+{\vec{M}}_{rt}^{b}+{\vec{M}}_{lt}^{b}+{\vec{M}}_{bt}^{b} \end{array}$$
where ${\vec{M}}_{r}^{b}$, ${\vec{M}}_{l}^{b}$, and ${\vec{M}}_{b}^{b}$ are the torque vectors generated by the rotor thrust, ${\vec{M}}_{rt}^{b}$, ${\vec{M}}_{lt}^{b}$, and ${\vec{M}}_{bt}^{b}$ represent the torque vectors caused by the rotor air resistance, and $Q_{r}$, $Q_{l}$, and $Q_{b}$ indicate the torques of the propeller.

According to the above analyses, Equation (2) is decomposed in the BBF coordinate frame and expressed as follows in order to describe the kinetic model more clearly:
(5)$$\begin{array}{l} \left\{\begin{array}{l} \left[\begin{array}{c} \dot{u}\\ \dot{v}\\ \dot{w} \end{array}\right]=\frac{1}{m}\left[\begin{array}{c} F_{x}{}^{b}\\ F_{y}{}^{b}\\ F_{z}{}^{b} \end{array}\right]+\left[\begin{array}{c} rv-qw\\ pw-ru\\ qu-pv \end{array}\right]\\ \left[\begin{array}{c} \dot{p}\\ \dot{q}\\ \dot{r} \end{array}\right]=\left[\begin{array}{c} \frac{1}{I_{xx}}\\ \frac{1}{I_{yy}}\\ \frac{1}{I_{zz}} \end{array}\right]\left[\begin{array}{c} M_{x}{}^{b}\\ M_{y}{}^{b}\\ M_{z}{}^{b} \end{array}\right]+\left[\begin{array}{c} \frac{\left(I_{yy}-I_{zz}\right)}{I_{xx}}qr\\ \frac{\left(I_{zz}-I_{xx}\right)}{I_{xx}}pr\\ \frac{\left(I_{xx}-I_{yy}\right)}{I_{xx}}pq \end{array}\right] \end{array}\right.\\ \left\{\begin{array}{l} \left[\begin{array}{c} {\dot{x}}^{n}\\ {\dot{y}}^{n}\\ {\dot{z}}^{n} \end{array}\right]=C_{b}^{n}\left[\begin{array}{c} u\\ v\\ w \end{array}\right]\\ \left[\begin{array}{c} \dot{\phi }\\ \dot{\theta }\\ \dot{\psi } \end{array}\right]=\left[\begin{array}{ccc} 1 & s\phi t\theta & s\phi t\theta \\ 0 & c\phi & -s\phi \\ 0 & s\phi /c\theta & c\phi /c\theta \end{array}\right]\left[\begin{array}{c} p\\ q\\ r \end{array}\right] \end{array}\right. \end{array}$$

## 3. Control Strategy

In this chapter, the control strategy of the tilt-rotor UAV in the VTOL phase is studied, and the attitude and position controllers are designed based on the dynamic model, respectively.

### 3.1. Control Schemes

#### 3.1.1. Conventional Control Scheme

When designing the flight control system of a tilt-rotor UAV in the VTOL phase, the conventional control scheme is to decouple the dynamic model into a fully actuated inner loop system and an underactuated outer loop system. The attitude subsystem is designed as an inner loop due to the wider frequency band and faster motion, while the position motion mode has a narrower frequency band, and the position’s subsystem is designed as an outer loop. Figure 3 depicts a block diagram of the conventional control scheme.

#### 3.1.2. Thrust Vectoring Control Scheme

The conventional control scheme works as follows: the controller gives the throttle command to the rotor motor, and the thrust moment, generated by the rotor, changes the attitude of the UAV, and the attitude and the total thrust work together to form lateral and forward forces on the UAV. The DTVC scheme works as follows: the controller gives the deflection command to the rotor servo, and the actuator responds quickly to generate direct lateral and forward thrusts. According to the above comparison analyses, the DTVC scheme has a faster response. The structure of the DTVC scheme is presented in Figure 4. The proposed scheme consists of a decoupled attitude hold loop and a position loop.

### 3.2. Controller Design

Based on the dynamics of the tilt-rotor UAV in the VTOL phase, the sliding mode controller combined with the backstepping method is designed for both the conventional and direct force control schemes. The state space model for the proposed tri-rotor mechanism can be written as follows, with the state vector, X, and the control vector, U:(6)$$\left\{\begin{array}{l} X={\left[\begin{array}{ccc} x_{1}, & \ldots & ,x_{12} \end{array}\right]}^{T}\\ \;={\left[\begin{array}{cccccccccccc} x & \dot{x} & y & \dot{y} & z & \dot{z} & \phi & \dot{\phi } & \theta & \dot{\theta } & \psi & \dot{\psi } \end{array}\right]}^{T}\\ U={\left[\begin{array}{ccc} u_{1}, & \ldots & ,u_{6} \end{array}\right]}^{T}\\ \;={\left[\begin{array}{cccccc} U_{x} & U_{y} & U_{z} & U_{\phi } & U_{\theta } & U_{\psi } \end{array}\right]}^{T} \end{array}\right.$$
where $U_{x}$, $U_{y}$, and $U_{z}$ represent the virtual control variables of the position channel, whose control equations can be calculated from Equation (3). Moreover, $U_{\phi }$, $U_{\theta }$, and $U_{\psi }$ denote the virtual control variables of the roll, pitch, and yaw channels, respectively, whose control equations can be calculated from Equation (10). To sum up, the total control equation of the UAV can be described as Equation (7):(7)$$\left[\begin{array}{c} U_{x}\\ U_{y}\\ U_{z}\\ U_{\phi }\\ U_{\theta }\\ U_{\psi } \end{array}\right]=\left[\begin{array}{c} {\vec{F}}_{x}^{b}\\ {\vec{F}}_{y}^{b}\\ {\vec{F}}_{z}^{b}\\ {\vec{M}}_{x}^{b}\\ {\vec{M}}_{y}^{b}\\ {\vec{M}}_{z}^{b} \end{array}\right]=\left[\begin{array}{c} \left(c\theta c\psi \right)\left(T_{rm}\cdot sA_{r}+T_{lm}\cdot sA_{l}\right)+\left(s\theta c\psi s\phi -s\psi c\phi \right)\left(T_{bm}\cdot sA_{b}\right)-\left(s\theta c\psi c\phi +s\psi s\phi \right)\left(T_{rm}\cdot cA_{r}+T_{lm}\cdot cA_{l}+T_{bm}\cdot cA_{l}\right)\\ \left(c\theta s\psi \right)\left(T_{rm}\cdot sA_{r}+T_{lm}\cdot sA_{l}\right)+\left(s\theta s\psi s\phi +c\psi c\phi \right)\left(T_{bm}\cdot sA_{b}\right)-\left(s\theta s\psi c\phi -c\psi s\phi \right)\left(T_{rm}\cdot cA_{r}+T_{lm}\cdot cA_{l}+T_{bm}\cdot cA_{l}\right)\\ \left(-s\theta \right)\left(T_{rm}\cdot sA_{r}+T_{lm}\cdot sA_{l}\right)+\left(c\theta s\phi \right)\left(T_{bm}\cdot sA_{l}\right)-\left(c\theta c\phi \right)\left(T_{rm}\cdot cA_{r}+T_{lm}\cdot cA_{l}+T_{bm}\cdot cA_{l}\right)+mg\\ T_{lm}\cdot cA_{l}\cdot y_{l}-T_{rm}\cdot cA_{r}\cdot y_{r}+Q_{r}\cdot sA_{r}-Q_{l}\cdot sA_{l}\\ T_{lm}\cdot cA_{l}\cdot x_{l}+T_{rm}\cdot cA_{r}\cdot x_{r}-T_{bm}\cdot cA_{b}\cdot x_{b}+Q_{b}\cdot sA_{b}\\ T_{lm}\cdot sA_{l}\cdot y_{l}-T_{rm}\cdot sA_{r}\cdot y_{r}-T_{bm}\cdot sA_{b}\cdot x_{b}-Q_{r}\cdot cA_{r}+Q_{l}\cdot cA_{l}-Q_{b}\cdot cA_{b} \end{array}\right]$$

In addition, in the VTOL phase, the attitude angle of the UAV changes very little, and the gyroscopic effect of the rotor can be ignored. In order to make the controller design simple, the following assumptions are made in Equation (8):(8)$$\left\{\begin{array}{l} \sin\phi =\phi ,\sin\theta =\theta ,\sin\psi =\psi \\ \cos\phi =1,\cos\theta =1,\cos\psi =1\\ \dot{\phi }=p,\dot{\theta }=q,\dot{\psi }=r\\ Q_{r}=Q_{l}=Q_{b}=0 \end{array}\right.$$

For the conventional control scheme, the tilt angles of the forward rotors are kept in the vertical state. Additionally, the rear motor keeps a small tilt angle. Therefore, the control model can be simplified as Equation (9):(9)$$\left[\begin{array}{c} U_{x}\\ U_{y}\\ U_{z}\\ U_{\phi }\\ U_{\theta }\\ U_{\psi } \end{array}\right]=\left[\begin{array}{c} \left(\theta \phi -\psi \right)\left(T_{bm}\cdot A_{b}\right)-\left(\theta +\psi \phi \right)\left(T_{rm}+T_{lm}+T_{bm}\right)\\ \left(\theta \psi \phi +1\right)\left(T_{bm}\cdot A_{b}\right)-\left(\theta \psi -\phi \right)\left(T_{rm}+T_{lm}+T_{bm}\right)\\ \left(\phi \right)\left(T_{bm}\cdot A_{b}\right)-\left(T_{rm}+T_{lm}+T_{bm}\right)+mg\\ T_{lm}\cdot y_{l}-T_{rm}\cdot y_{r}\\ T_{lm}\cdot x_{l}+T_{rm}\cdot x_{r}-T_{bm}\cdot x_{b}\\ -T_{bm}\cdot A_{b}\cdot x_{b} \end{array}\right]$$

Since the UAV does not maneuver in the VTOL phase, the attitude loop of the UAV is controlled to increase the stability of the DTVC scheme. Therefore, the tilt angles of the three rotors both keep a small angle. When crossing and different conditions occur, the horizontal position of the UAV deviates. To address this issue, the horizontal position control is investigated in this scheme, which is mainly used to maintain an accurate and stable position control without a large attitude maneuver control so that the UAV can land and take-off smoothly. To sum up, the control model can be simplified as Equation (10):(10)$$\left[\begin{array}{c} U_{x}\\ U_{y}\\ U_{z}\\ U_{\phi }\\ U_{\theta }\\ U_{\psi } \end{array}\right]=\left[\begin{array}{c} \left(T_{rm}\cdot A_{r}+T_{lm}\cdot A_{l}\right)+\left(\theta \phi -\psi \right)\left(T_{bm}\cdot A_{b}\right)-\left(s\theta +\psi \phi \right)\left(T_{rm}+T_{lm}+T_{bm}\right)\\ \psi \left(T_{rm}\cdot A_{r}+T_{lm}\cdot A_{l}\right)+\left(\theta \psi \phi +1\right)\left(T_{bm}\cdot A_{b}\right)-\left(\theta \psi -\phi \right)\left(T_{rm}+T_{lm}+T_{bm}\right)\\ \left(-\theta \right)\left(T_{rm}\cdot A_{r}+T_{lm}\cdot A_{l}\right)+\phi \left(T_{bm}\cdot A_{b}\right)-\left(T_{rm}+T_{lm}+T_{bm}\right)+mg\\ T_{lm}\cdot y_{l}-T_{rm}\cdot y_{r}\\ T_{lm}\cdot x_{l}+T_{rm}\cdot x_{r}-T_{bm}\cdot x_{b}\\ T_{lm}\cdot A_{l}\cdot y_{l}-T_{rm}\cdot A_{r}\cdot y_{r} \end{array}\right]$$

In the simulation test, Equation (9) represents the control equation of the conventional control scheme, and Equation (10) represents the control equation of the direct thrust vectoring control scheme. In addition, the control allocation strategy of the two schemes in Table 1 is also obtained according to Equations (9) and (10).

Moreover, the UAV’s state-space model is given by:(11)$$\begin{array}{l} f\left(X,U\right)=\left\{\begin{array}{l} x_{2}={\dot{x}}_{1}\\ {\dot{x}}_{2}=a_{1}\cdot U_{x}+d_{x}\left(t\right)\\ x_{4}={\dot{x}}_{3}\\ {\dot{x}}_{4}=a_{2}\cdot U_{y}+d_{y}\left(t\right)\\ x_{6}={\dot{x}}_{5}\\ {\dot{x}}_{6}=a_{3}\cdot U_{z}+d_{z}\left(t\right)\\ x_{8}={\dot{x}}_{7}\\ {\dot{x}}_{8}=a_{4}\cdot U_{\phi }+b_{1}\cdot x_{10}\cdot x_{12}+d_{\phi }\left(t\right)\\ x_{10}={\dot{x}}_{9}\\ {\dot{x}}_{10}=a_{5}\cdot U_{\theta }+b_{2}\cdot x_{8}\cdot x_{12}+d_{\theta }\left(t\right)\\ x_{12}={\dot{x}}_{11}\\ {\dot{x}}_{12}=a_{6}\cdot U_{\psi }+b_{3}\cdot x_{8}\cdot x_{10}+d_{\psi }\left(t\right) \end{array}\right. \end{array}$$
where $a_{1}=a_{2}=a_{3}=1/m$, $a_{4}=\left(I_{yy}-I_{zz}\right)/\;I_{xx}$, $a_{5}=\left(I_{zz}-I_{xx}\right)/\;I_{yy}$, $a_{6}=\left(I_{xx}-I_{yy}\right)/\;I_{zz}$, and $b_{1}=1/\;I_{xx}$, $b_{2}=1/\;I_{yy}$, $b_{3}=1/\;I_{zz}$. In this section, we take the roll control loop as an example for the stability analysis, and the different control loops, including the pitch, yaw, forward, lateral, and vertical loops, are similar. The roll states’ subsystem in Equation (11) are considered as follows:(12)$$\left\{\begin{array}{l} x_{8}={\dot{x}}_{7}\\ {\dot{x}}_{8}=a_{4}\cdot U_{\phi }+b_{1}\cdot x_{10}\cdot x_{12}+d_{\phi }\left(t\right) \end{array}\right.$$
where $d_{\phi }\left(t\right)$ is the bounded interference term of the model, and $\left|d_{\phi }\left(t\right)\right|<D$.

Assuming that the target value of the roll angle in the roll control loop is $\phi_{c}=x_{7c}$, then the tracking error variable, ${\dot{e}}_{\phi }$, and its derivative, $e_{\phi }$, are expressed as follows:(13)$$\left\{\begin{array}{l} e_{\phi }=x_{7}-x_{7c}\\ {\dot{e}}_{\phi }={\dot{x}}_{7}-{\dot{x}}_{7c}=x_{8}-{\dot{x}}_{7c} \end{array}\right.$$

Define the Lyapunov function and its derivative as follows:(14)$$\left\{\begin{array}{l} V_{1}=1/2e_{\phi }^{2}\\ {\dot{V}}_{1}=e_{\phi }\cdot {\dot{e}}_{\phi }=e_{\phi }\left(x_{8}-{\dot{x}}_{7c}\right) \end{array}\right.$$

Then, define the virtual control variable, $\alpha_{4}$, as follows:(15)$$\left\{\begin{array}{l} \alpha_{4}=x_{8}+c_{4}\cdot e_{\phi }-{\dot{x}}_{7c}\\ c_{4}>0 \end{array}\right.$$

As in Equation (15), the derivative of the Lyapunov function may be computed as:(16)$${\dot{V}}_{1}=-c_{4}\cdot e_{\phi }^{2}+e_{\phi }\cdot \alpha_{4}$$

The system is unstable because of the term $e_{\phi }\cdot \alpha_{4}$ in Equation (16). In combination with the sliding mode variable structure control method, the sliding surface and the Lyapunov function are defined as:(17)$$\left\{\begin{array}{l} s_{\phi }=\alpha_{4}\\ V_{2}=V_{1}+1/2\alpha_{4}^{2} \end{array}\right.$$

The derivative of $s_{\phi }$ and $V_{2}$ in Equation (17) is expressed as:(18)$$\left\{\begin{array}{l} {\dot{s}}_{\phi }=b_{1}\cdot x_{10}\cdot x_{12}+a_{4}\cdot U_{\phi }+c_{4}\cdot {\dot{e}}_{\phi }-{\ddot{x}}_{7c}+d_{\phi }\left(t\right)\\ {\dot{V}}_{2}={\dot{V}}_{1}+s_{\phi }\cdot {\dot{s}}_{\phi }\\ \;\;\;\;\;=-c_{4}\cdot e_{\phi }^{2}+e_{\phi }\cdot \alpha_{4}+\alpha_{4}\left(b_{1}\cdot x_{10}\cdot x_{12}+a_{4}\cdot U_{\phi }+c_{4}\cdot {\dot{e}}_{\phi }-{\ddot{x}}_{7c}+d_{\phi }\left(t\right)\right) \end{array}\right.$$

Considering Equation (12), the derivative of the sliding surfaces in Equation (18) becomes:(19)$${\dot{s}}_{\phi }=-\varepsilon_{4}\mathrm{sgn}\left(s_{\phi }\right)-k_{4}\cdot s_{\phi }$$
where the control law coefficient is $\varepsilon_{4}>1,k_{4}>0$.

In order to satisfy ${\dot{V}}_{2}\leq 0$, we may obtain the following equivalent control input in the roll loop:(20)$$U_{\phi }=\frac{1}{a_{4}}\left[-\varepsilon_{4}\mathrm{sgn}\left(\alpha_{4}\right)-b_{1}\cdot x_{10}\cdot x_{12}-k_{4}\cdot \alpha_{4}+{\ddot{x}}_{7c}+c_{4}\cdot {\dot{e}}_{\phi }-e_{\phi }\right]$$

The stability proof is: (21)$$\begin{array}{l} {\dot{V}}_{2}=-c_{4}\cdot e_{\phi }^{2}-k_{4}\cdot \alpha_{4}^{2}-\varepsilon_{4}\cdot \left|\alpha_{4}\right|+\alpha_{4}\cdot d_{\phi }\left(t\right)\\ \;\;\;\;\;\leq -c_{4}\cdot e_{\phi }^{2}-k_{4}\cdot \alpha_{4}^{2}\leq 0 \end{array}$$

According to Lyapunov’s stability theorem, the system can be asymptotically stable. As is analyzed above, the dynamic sliding mode functions are depicted as follows:(22)$$\left\{\begin{array}{l} s_{x}=c_{1}e_{x}+{\dot{e}}_{x}\\ s_{y}=c_{2}e_{y}+{\dot{e}}_{y}\\ s_{z}=c_{3}e_{z}+{\dot{e}}_{z}\\ s_{\phi }=c_{4}e_{\phi }+{\dot{e}}_{\phi }\\ s_{\theta }=c_{5}e_{\theta }+{\dot{e}}_{\theta }\\ s_{\psi }=c_{6}e_{\psi }+{\dot{e}}_{\psi } \end{array}\right.$$
where $e_{x}=x_{c}-x$, $e_{y}=y_{c}-y$, $e_{z}=z_{c}-z$, $e_{z}=z_{c}-z$, $e_{\theta }=\theta_{c}-\theta$,$e_{\phi }=\phi_{c}-\phi$, and $e_{\psi }=\psi_{c}-\psi$.

Then, the stabilizing control laws are as follows:(23)$$\left\{\begin{array}{l} U_{x}=\frac{1}{a_{1}}\left[-\varepsilon_{1}\mathrm{sgn}\left(\alpha_{1}\right)-k_{1}\cdot \alpha_{1}+{\ddot{x}}_{1c}+c_{1}\cdot {\dot{e}}_{x}-e_{x}\right]\\ U_{y}=\frac{1}{a_{2}}\left[-\varepsilon_{2}\mathrm{sgn}\left(\alpha_{2}\right)-k_{2}\cdot \alpha_{2}+{\ddot{x}}_{3c}+c_{2}\cdot {\dot{e}}_{y}-e_{y}\right]\\ U_{z}=\frac{1}{a_{3}}\left[-\varepsilon_{3}\mathrm{sgn}\left(\alpha_{3}\right)-k_{3}\cdot \alpha_{3}+{\ddot{x}}_{5c}+c_{3}\cdot {\dot{e}}_{z}-e_{z}\right]\\ U_{\phi }=\frac{1}{a_{4}}\left[-\varepsilon_{4}\mathrm{sgn}\left(\alpha_{4}\right)-b_{1}\cdot x_{10}\cdot x_{12}-k_{4}\cdot \alpha_{4}+{\ddot{x}}_{7c}+c_{4}\cdot {\dot{e}}_{\phi }-e_{\phi }\right]\\ U_{\theta }=\frac{1}{a_{5}}\left[-\varepsilon_{5}\mathrm{sgn}\left(\alpha_{5}\right)-b_{2}\cdot x_{8}\cdot x_{12}-k_{5}\cdot \alpha_{5}+{\ddot{x}}_{9c}+c_{5}\cdot {\dot{e}}_{\theta }-e_{\theta }\right]\\ U_{\psi }=\frac{1}{a_{6}}\left[-\varepsilon_{6}\mathrm{sgn}\left(\alpha_{6}\right)-b_{3}\cdot x_{8}\cdot x_{10}-k_{6}\cdot \alpha_{6}+{\ddot{x}}_{11c}+c_{6}\cdot {\dot{e}}_{\psi }-e_{\psi }\right] \end{array}\right.$$

### 3.3. Control Allocation

The novel tilt-rotor UAV studied in this paper has a relatively large number of actuators, including three motors with three tilt mechanisms with the respective control variables $T_{rm}$, $T_{lm}$, $T_{bm}$, $A_{r}$, $A_{l}$, and $A_{b}$.

In this paper, the position control and attitude control are decoupled, and the UAV is assumed to be in equilibrium for the control allocation. Therefore, the main rotor thrust is $T_{rm}\approx T_{lm}$, and the body attitude angle is ϕ=θ=ψ=0. According to Equation (10), the forward channel control variable, $U_{x}$, the lateral channel control variable, $U_{y}$, and the vertical channel control variable, $U_{z}$, can be simplified as follows:(24)$$\begin{array}{l} \left[\begin{array}{c} U_{x}\\ U_{y}\\ U_{z} \end{array}\right]=\left[\begin{array}{c} \left(T_{rm}\cdot A_{r}+T_{lm}\cdot A_{l}\right)+\left(\theta \phi -\psi \right)\left(T_{bm}\cdot A_{b}\right)-\left(\theta +\psi \phi \right)\left(T_{rm}+T_{lm}+T_{bm}\right)\\ \psi \left(T_{rm}\cdot A_{r}+T_{lm}\cdot A_{l}\right)+\left(\theta \psi \phi +1\right)\left(T_{bm}\cdot A_{b}\right)-\left(\theta \psi -\phi \right)\left(T_{rm}+T_{lm}+T_{bm}\right)\\ \left(-\theta \right)\left(T_{rm}\cdot A_{r}+T_{lm}\cdot A_{l}\right)+\phi \left(T_{bm}\cdot A_{b}\right)-\left(T_{rm}+T_{lm}+T_{bm}\right)+mg \end{array}\right]\\ \;\;\;\;\;\;\;\;\;\;\;\;\approx \left[\begin{array}{c} T_{rm}\cdot \left(A_{r}+A_{l}\right)\\ T_{bm}\cdot \left(A_{b}\right)\\ \left(T_{rm}+T_{lm}+T_{bm}\right)+mg \end{array}\right] \end{array}$$

Based on the model parameters in Table 2 and Equation (10), the roll channel control variable, $U_{\phi }$, pitch channel control variable, $U_{\theta }$, and yaw channel control variable, $U_{\psi }$, can be simplified as follows:(25)$$\begin{array}{l} \left[\begin{array}{c} U_{\phi }\\ U_{\theta }\\ U_{\psi } \end{array}\right]=\left[\begin{array}{c} T_{lm}\cdot y_{l}-T_{rm}\cdot y_{r}\\ T_{lm}\cdot x_{l}+T_{rm}\cdot x_{r}-T_{bm}\cdot x_{b}\\ T_{lm}\cdot A_{l}\cdot y_{l}-T_{rm}\cdot A_{r}\cdot y_{r} \end{array}\right]\\ \;\;\;\;\;\;\;\;\;\;\;\;\approx \left[\begin{array}{c} y_{l}\left(T_{lm}-T_{rm}\right)\\ x_{l}\left(T_{lm}+T_{rm}\right)-T_{bm}\cdot x_{b}\\ T_{lm}\cdot y_{l}\left(A_{l}-A_{r}\right) \end{array}\right] \end{array}$$

According to the above analysis, when the position channel of the UAV is decoupled from the attitude channel, its forward control channel variable, $U_{x}$, can be allocated to the co-deflection of the main rotor’s tilt angle separately, the lateral channel control variable, $U_{y}$, can be allocated to the tilting of the back rotor, the vertical channel control variable, $U_{z}$, can be allocated to the total thrust, the roll channel control variable, $U_{\phi }$, can be allocated to the main rotor thrusts’ differential, the pitch channel control variable, $U_{\theta }$, can be allocated to the main rotor and tail rotor thrusts’ differential, and the yaw channel control variable, $U_{\psi }$, can be allocated to the main rotor tilt angle differential.

Therefore, the control allocation strategy of the actuation structure for different control schemes in VTOL phase can be expressed as Table 1.

**Table 1 sensors-23-00574-t001:** Control allocation for different schemes in the VTOL phase.

Control Channel	Conventional Scheme	Thrust Vectoring Scheme
Roll control $U_{\phi }$	$T_{n}$	$T_{n}$
Pitch control $U_{\theta }$	$2x_{r}\cdot T_{m}-x_{b}\cdot T_{bm}$	$2x_{r}\cdot T_{m}-x_{b}\cdot T_{bm}$
Yaw control $U_{\psi }$	$A_{b}$	$A_{n}$
Forward control $U_{x}$	Pitching movement	$A_{m}$
Lateral control $U_{y}$	Rolling movement	$A_{b}$
Vertical control $U_{z}$	$2T_{m}+T_{bm}$	$2T_{m}+T_{bm}$

Where $T_{m}$ is the main motor thrusts’ synchronization, $T_{n}$ is the main motor thrusts’ differential, $A_{m}$ is the main rotor tilt angle synchronization, and $A_{n}$ is the main rotor tilt angle differential, and they can be defined as
(26)$$\left\{\begin{array}{l} T_{m}=\frac{1}{2}(T_{rm}+T_{lm})\\ T_{n}=\frac{1}{2}(T_{rm}-T_{lm})\\ A_{m}=\frac{1}{2}(A_{r}+A_{l})\\ A_{n}=\frac{1}{2}(A_{r}-A_{l}) \end{array}\right.$$

#### 3.3.1. Control Allocation Strategy of Conventional Scheme

As shown in Figure 5, the attitude channel’s control strategy is presented to differentially adjust the propeller speeds of the main rotors on each side of the fuselage to generate the roll torque in the roll control channel. The pitch torque is obtained by the differential tuning of the propeller speeds of the rotors at the front and rear of the fuselage, and the yaw torque is obtained by adjusting the tilt angle of the rear rotor.

#### 3.3.2. Control Allocation Strategy of Thrust Vectoring Scheme

As shown in Figure 6, the control strategies for the roll, pitch, and droop channels are the same as the conventional control scheme, while the yaw channel control’s strategy is to differentially adjust the tilt angles of the left and right main rotors of the fuselage to obtain the yaw torque and finally maintain the desired attitude angle. Moreover, the rotors’ tilting mechanism and the corresponding authority of the direct control of the rotor thrust leads to the UAV having full actuation. On the one hand, the UAV can manipulate its forward projection of the rotor thrust vector rather than pitch its body, as is typical for the control strategies of common multirotor platforms. On the other hand, the UAV can obtain lateral thrust vector projection by tilting the tail motor without the necessity of rolling its body.

## 4. Simulations

In this section, the simulation results are presented in order to observe the effectiveness of the derived model and the performances of the proposed backstepping sliding mode controller, and the actual control variables allocated by different control schemes. The simulation results are based on the following real parameters of the tri-rotor depicted in Figure 2. The detailed model parameters of the novel tilt-rotor UAV are specified, as shown in Table 2.

The control laws, $U_{i\left(i=\phi \ldots z\right)}$, have been calculated, and the actual control variables can be allocated in Section 3.

**Table 2 sensors-23-00574-t002:** The model parameters of the novel tilt-rotor UAV.

Parameter	Definition	Value	Unit
m	Mass	70	kg
$I_{xx}$	Roll inertia	43.91	kg·m^2^
$I_{yy}$	Pitch inertia	15.13	kg·m^2^
$I_{zz}$	Yaw inertia	57.21	kg·m^2^
b	Wingspan	3.15	m
$S_{w}$	Wing aera	4.01	m^2^
$c_{A}$	Chord length	1.27	m
${\left(x_{r},y_{r},z_{r}\right)}^{T}$	Right rotor position	${\left(0.05,1.75,0.03\right)}^{T}$	m
${\left(x_{l},y_{l},z_{l}\right)}^{T}$	Left rotor position	${\left(0.05,-1.75,0.03\right)}^{T}$	m
${\left(x_{b},y_{b},z_{b}\right)}^{T}$	Back rotor position	${\left(-0.85,0,0.08\right)}^{T}$	m

### 4.1. Simulation of Backstepping Sliding Mode Controller

Taking the roll loop controller, for example, we can see the control effects with the backstepping sliding mode controller. The desired angle is chosen as $\phi_{c}\left(t\right)=\sin\left(t\right)$, the initial states are 0.2, and the control law adopts Equation (20). The simulation results are shown in Figure 7.

The simulation results in Figure 7 show the following conclusions:
In terms of the system response time, the roll angle tracking control has a response time of 1.9 s, which is excellent for the dynamic response;In terms of control accuracy, the controller maintains high accuracy and stable tracking of the desired angle.

In summary, the proposed backstepping-based controller combined with the sliding mode variational control algorithm can well meet the requirements of the control response speed, accuracy, and stability of the system.

### 4.2. Simulation of Different Control Schemes

#### 4.2.1. Simulation of Target Point Flight

Neglecting the effect of wind disturbance, the DTVC scheme and the conventional scheme are used to perform a comparative simulation of the track target point flight to determine the difference in the control accuracy and the ability of the response speed between the different control schemes.

The initial position is ${\left(X_{0},Y_{0},Z_{0}\right)}^{T}={\left(0,0,0\right)}^{T}\left(m\right)$, the initial velocity is ${\left(V_{x0},V_{y0},V_{z0}\right)}^{T}={\left(0,0,0\right)}^{T}\left(m/s\right)$, the initial attitude angle is ${\left(\phi_{0},\theta_{0},\psi_{0}\right)}^{T}={\left(0^{\circ },0^{\circ },0^{\circ }\right)}^{T}$, the initial attitude angle velocity is ${\left({\dot{\phi }}_{0},{\dot{\theta }}_{0},{\dot{\psi }}_{0}\right)}^{T}={\left(0,0,0\right)}^{T}\left({}^{\circ }/s\right)$, the desired position is ${\left(X_{f},Y_{f},Z_{f}\right)}^{T}={\left(15,-15,15\right)}^{T}\left(m\right)$, the desired velocity is ${\left(V_{xf},V_{yf},V_{zf}\right)}^{T}={\left(0,0,0\right)}^{T}\left(m/s\right)$, the desired attitude angle is ${\left(\phi_{f},\theta_{f},\psi_{f}\right)}^{T}={\left(0^{\circ },0^{\circ },0^{\circ }\right)}^{T}$, and the desired attitude angle velocity is ${\left({\dot{\phi }}_{f},{\dot{\theta }}_{f},{\dot{\psi }}_{f}\right)}^{T}={\left(0,0,0\right)}^{T}\left({}^{\circ }/s\right)$. The simulation results are shown in Figure 8, Figure 9 and Figure 10.

Figure 8 shows the comparison curves of the UAV’s attitude angle and the attitude angular velocity for the two schemes, respectively. As shown in the simulation results, the DTVC scheme can modify the flight path of the UAV while keeping the attitude angle essentially unchanged without the attitude angle overshooting negatively, as in the conventional control scheme.

Figure 9 shows a comparison of the response curves for the UAV’s position and velocity when the two schemes are employed separately. The simulation results show that the DTVC scheme can quickly track the trajectory command and has a higher response speed than the conventional control scheme.

Figure 10 shows the response curve of the control’s variation of the UAV’s actuator. According to the simulation results, the rotor tilt angle and rotor tension of the UAV are shifted more gently in the DTVC scheme compared to the conventional control scheme, and the actuator effectiveness is fully applied. In this paper, we demonstrate the effectiveness of the DTVC scheme in the UAV’s control.

#### 4.2.2. Simulation of Fixed Height Flight with Attitude Disturbance

Considering the effect of attitude disturbance, the DTVC scheme and the conventional scheme are used to perform a comparative simulation of the flight in fixed height to determine the difference in the variation trend of the control actuators and the control ability of the response speed between the two control schemes.

The initial position is ${\left(X_{0},Y_{0},Z_{0}\right)}^{T}={\left(0,0,15\right)}^{T}\left(m\right)$, the initial velocity is ${\left(V_{x0},V_{y0},V_{z0}\right)}^{T}={\left(0,0,0\right)}^{T}\left(m/s\right)$, the initial attitude angle is ${\left(\phi_{0},\theta_{0},\psi_{0}\right)}^{T}={\left(5^{\circ },5^{\circ },0^{\circ }\right)}^{T}$, the initial attitude angle velocity is ${\left({\dot{\phi }}_{0},{\dot{\theta }}_{0},{\dot{\psi }}_{0}\right)}^{T}={\left(0,0,0\right)}^{T}\left({}^{\circ }/s\right)$, the desired position is ${\left(X_{f},Y_{f},Z_{f}\right)}^{T}={\left(15,-15,15\right)}^{T}\left(m\right)$, the desired velocity is ${\left(V_{xf},V_{yf},V_{zf}\right)}^{T}={\left(0,0,0\right)}^{T}\left(m/s\right)$, the desired attitude angle is ${\left(\phi_{f},\theta_{f},\psi_{f}\right)}^{T}={\left(0^{\circ },0^{\circ },0^{\circ }\right)}^{T}$, and the desired attitude angle velocity is ${\left({\dot{\phi }}_{f},{\dot{\theta }}_{f},{\dot{\psi }}_{f}\right)}^{T}={\left(0,0,0\right)}^{T}\left({}^{\circ }/s\right)$. The simulation results are shown in Figure 11, Figure 12 and Figure 13.

Figure 11 shows the comparison curves of the UAV’s attitude angle and the attitude angular velocity for the two schemes. Figure 12 shows a comparison of the response curves for the UAV position and velocity. As shown in the simulation results, the DTVC scheme can decouple the attitude angle control loop from the position control loop so that the attitude angle of the UAV converges more quickly and robustly compared to the conventional scheme. As a result, there is no large position control overshoot due to the small shift of the attitude angle. In addition, the DTVC scheme can also quickly track the position commands and has a higher trajectory response speed.

Figure 13 shows the response curve of the variation of the UAV’s actuator control. The simulation results show that the UAV with the DTVC scheme for the fixed height flight control simulation has a minor variation in the actuators and inferior correction frequency compared to the conventional control scheme.

#### 4.2.3. Simulation of Landing with Crosswind Disturbance

Considering the effect of crosswind disturbance, the direct thrust vector scheme and the conventional scheme are used to perform a comparative simulation of the landing to determine the difference in the control performance.

In order to achieve an accurate automatic landing, the UAV needs position control during the landing phase. When the conventional control scheme is adopted for landing, if there is a disturbance force from the side of the aircraft, the body will generate a roll angle to counteract the disturbance. Due to the high aspect ratio of the UAV and the layout characteristics of the wing body fusion, this can lead to the problem of the wing tip or paddle tip touching the ground.

Thus, in this paper, the crosswind disturbance refers to the interference from the side wind to the aircraft. In the simulation study, the crosswind disturbance is simplified as an additional sideways interference force attached to the UAV model. In this simulation test, Equation (3) can be revised as:(27)$${\vec{F}}^{b}={\vec{T}}_{r}^{b}+{\vec{T}}_{l}^{b}+{\vec{T}}_{b}^{b}+{\vec{G}}^{b}+{\vec{F}}_{w}^{b}$$
where ${\vec{F}}_{w}^{b}$ is the force of the crosswind disturbance.

The initial position is ${\left(X_{0},Y_{0},Z_{0}\right)}^{T}={\left(15,-15,15\right)}^{T}\left(m\right)$, the initial velocity is ${\left(V_{x0},V_{y0},V_{z0}\right)}^{T}={\left(0,0,0\right)}^{T}\left(m/s\right)$, the initial attitude angle is ${\left(\phi_{0},\theta_{0},\psi_{0}\right)}^{T}={\left(0^{\circ },0^{\circ },0^{\circ }\right)}^{T}$, the initial attitude angle velocity is ${\left({\dot{\phi }}_{0},{\dot{\theta }}_{0},{\dot{\psi }}_{0}\right)}^{T}={\left(0,0,0\right)}^{T}\left({}^{\circ }/s\right)$, the desired position is ${\left(X_{f},Y_{f},Z_{f}\right)}^{T}={\left(0,0,0\right)}^{T}\left(m\right)$, the desired velocity is ${\left(V_{xf},V_{yf},V_{zf}\right)}^{T}={\left(0,0,0\right)}^{T}\left(m/s\right)$, the desired attitude angle is ${\left(\phi_{f},\theta_{f},\psi_{f}\right)}^{T}={\left(0^{\circ },0^{\circ },0^{\circ }\right)}^{T}$, and the desired attitude angle velocity is ${\left({\dot{\phi }}_{f},{\dot{\theta }}_{f},{\dot{\psi }}_{f}\right)}^{T}={\left(0,0,0\right)}^{T}\left({}^{\circ }/s\right)$. The force of the crosswind disturbance is ${\vec{F}}_{w}^{b}=80\;\left(N\right)$. The simulation results are shown in Figure 14, Figure 15 and Figure 16.

Figure 14 shows the comparison of the response curves of the UAV’s attitude angle and attitude angular velocity for the crosswind landing simulation. Figure 15 shows a comparison of the position and velocity response curves of the UAV. Figure 16 shows the response curve of the control’s variation of the UAV’s actuator.

As shown in the simulation results, in the conventional scheme, the roll angle $\left(\phi =7.4^{\circ }\right)$ is larger when landing in crosswise winds compared with the DTVC scheme. Therefore, the wing tip will touch the ground when the landing gear is less than 0.7 m, according to the wingspan and the blade length of the UAV. However, the UAV with the DTVC scheme can maintain a stable attitude while landing in crosswind disturbance. Under the influence of crosswinds, the aircraft quickly achieves the UAV’s attitude stability with minor adjustments. The desired goal of an accurate landing with the UAV’s attitude stabilization is achieved.

#### 4.2.4. Monte Carlo Random Experiment of Landing Simulation

Due to the stochastic nature of crosswind disturbance, the crosswind disturbance is set to ${\vec{F}}_{w}^{b}=80+D\left[Rand\left(0\sim 1\right)\right]\left(N\right)$ in order to fit the actual situation and eliminate the contingencies. D is the boundary of the crosswind disturbance. In addition, according to the UAV’s structural parameters and simulation results, Equation (25) is the condition for a successful and precise landing. Monte Carlo simulation tests were performed for the two control schemes with 50 trials each.
(28)$$\left\{\begin{array}{l} t\in \left[0:12\right]\;\left(s\right)\\ \phi_{f}\in \left[-2:2\right]\;\left({}^{\circ }\right)\\ X_{n}\in \left[-1:1\right]\;\left(m\right)\\ Y_{n}\in \left[-1:1\right]\;\left(m\right) \end{array}\right.$$

Figure 17 shows a statistical plot of the landing locations for both sets of simulations. The dashed line is the boundary of the landing condition, and a UAV landing within 12 s is considered a successful precision landing if it lands within the dashed line and satisfies the attitude angle condition.

As shown in Figure 17, in the presence of large crosswind interference, the UAV landing sites are all concentrated within the boundary line with a 100% landing success rate and dense landing site distribution with the DTVC scheme. However, with the conventional control scheme, the landing sites are scattered, and the success rate is only 24 percent.

## 5. Conclusions

In this paper, the nonlinear mathematical model of a novel tilt-rotor UAV was derived. A controller based on the backstepping sliding mode control method was designed. Additionally, a direct thrust vectoring control (DTVC) scheme was proposed, and the simulation tests were established.

The simulation showed that the backstepping sliding mode controller could effectively control the roll angle, pitch angle, and yaw angle of the novel tilt-rotor UAV. The DTVC scheme improves the landing accuracy and speed compared to the CAPC scheme with attitude errors and crosswind disturbance and decouples the position control loop from the attitude control loop, avoiding the problem of the paddle tip touching the ground. What is more, with small errors and an overshoot of the UAV’s actuators, the scheme has desirable control performances. So, it can be concluded that the nonlinear model of the tilt-rotor UAV is valid, the backstepping sliding mode method is feasible in the controller design, and the DTVC scheme is more excellent in control performance compared with the conventional.

## Figures and Tables

**Figure 1 sensors-23-00574-f001:**
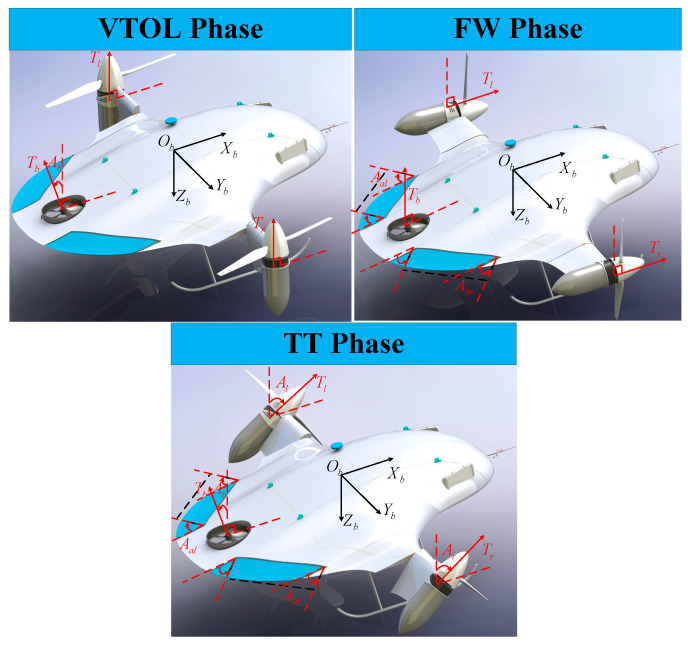
The flight phase of the novel tilt-rotor UAV.

**Figure 2 sensors-23-00574-f002:**
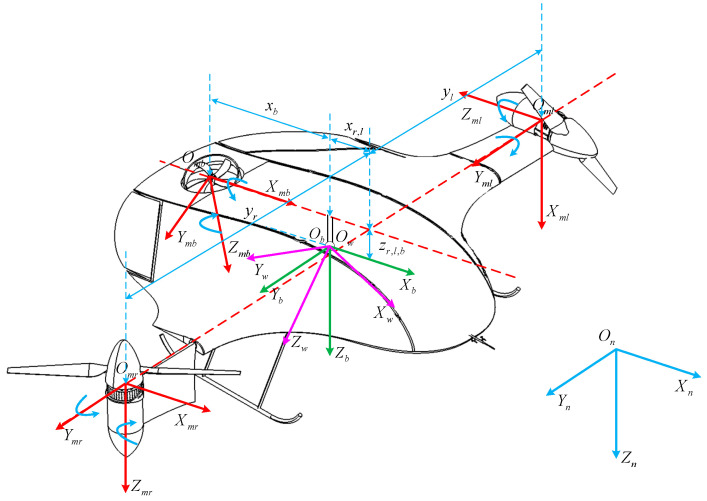
The schematic and frame of the novel tilt-rotor UAV.

**Figure 3 sensors-23-00574-f003:**
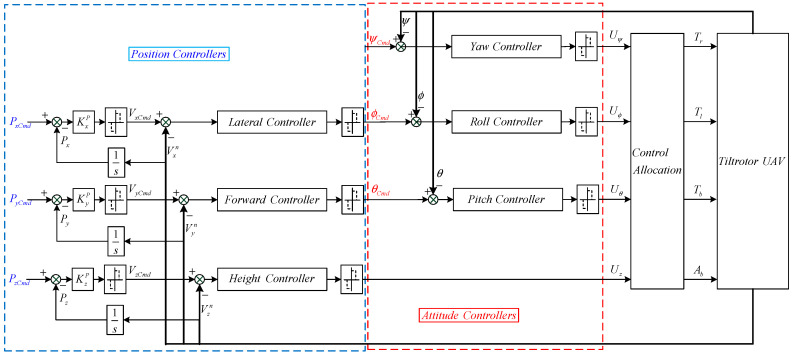
Structure diagram of conventional control scheme in the VTOL phase.

**Figure 4 sensors-23-00574-f004:**
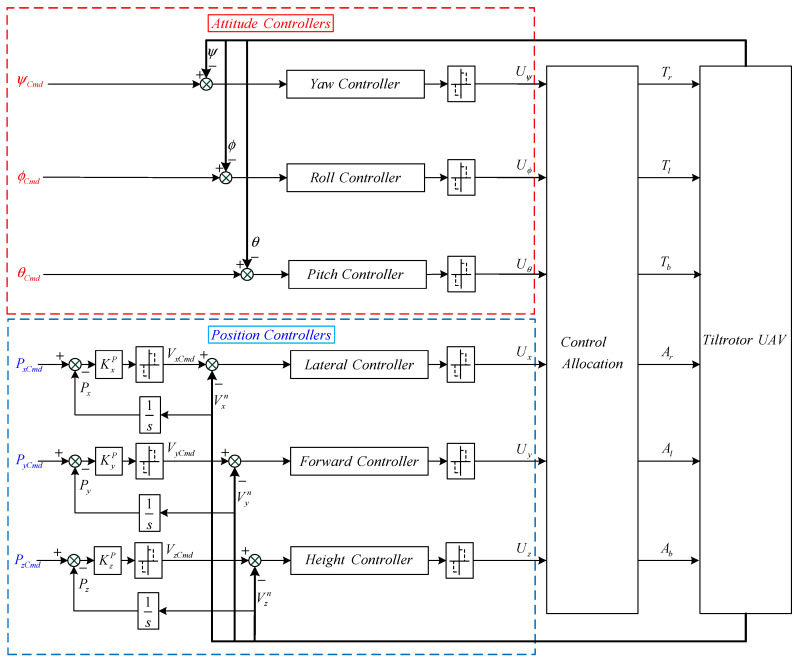
Structure diagram of the DTVC scheme in the VTOL phase.

**Figure 5 sensors-23-00574-f005:**
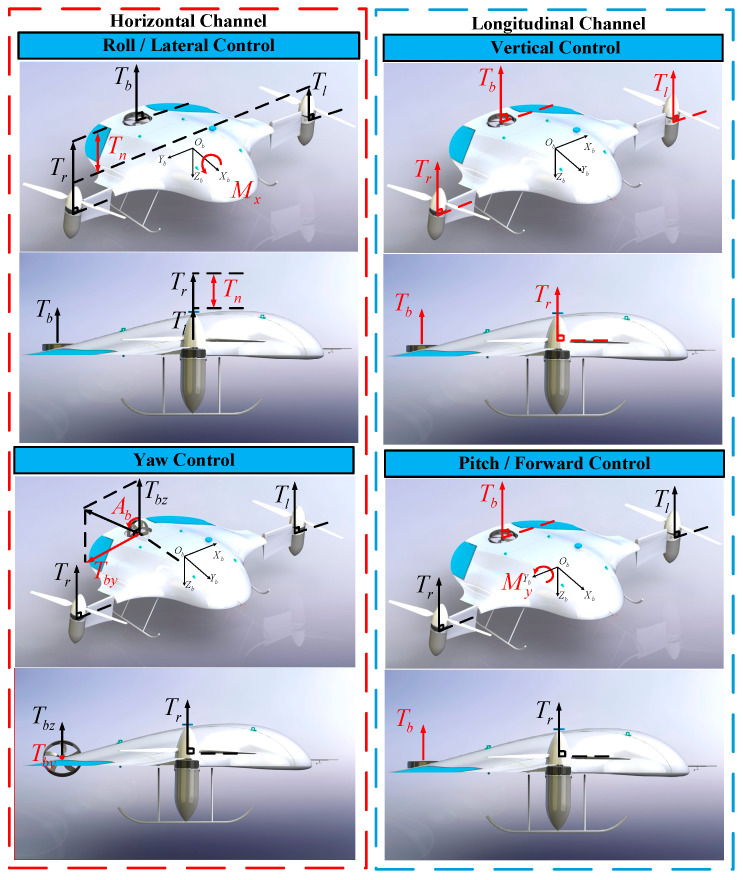
Control allocation strategy for the conventional scheme.

**Figure 6 sensors-23-00574-f006:**
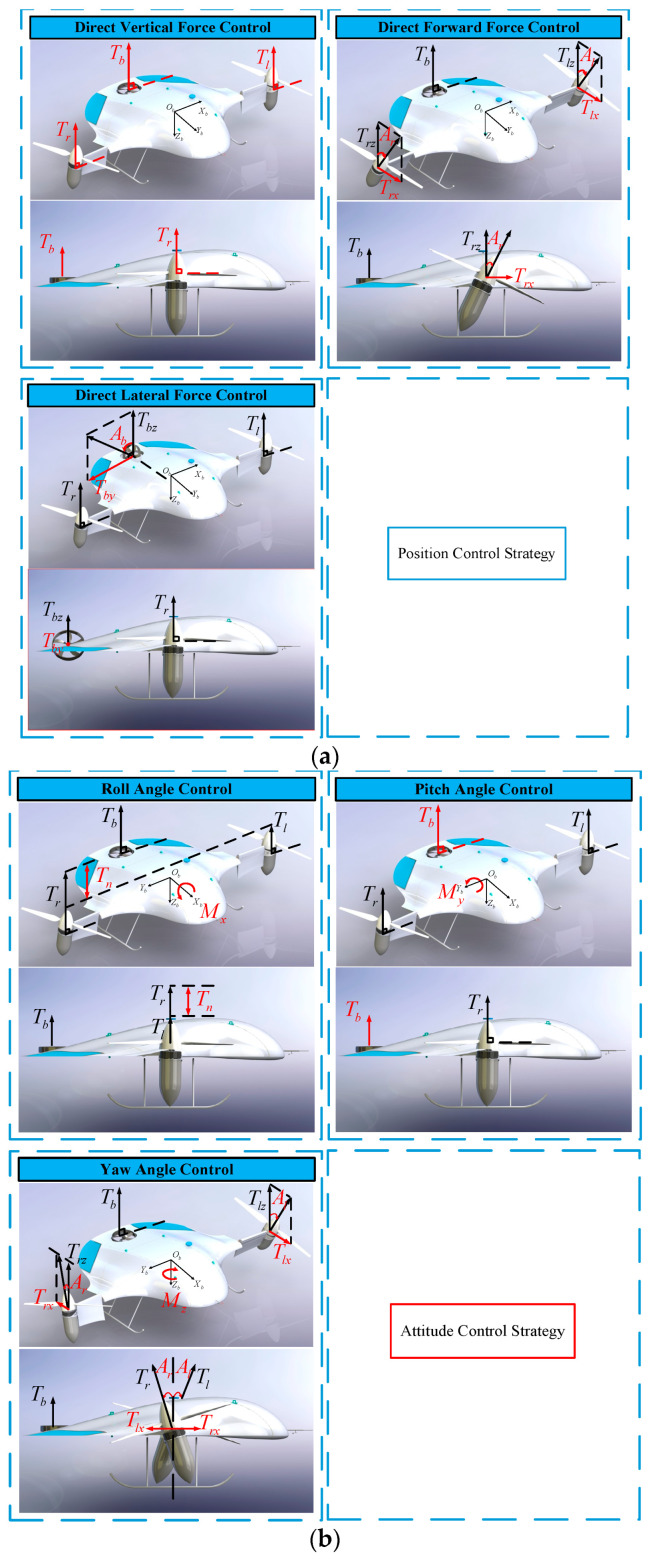
Control allocation strategy in the DTVC scheme: (**a**) control allocation strategy for the position channel; (**b**) control allocation strategy for the attitude channel.

**Figure 7 sensors-23-00574-f007:**
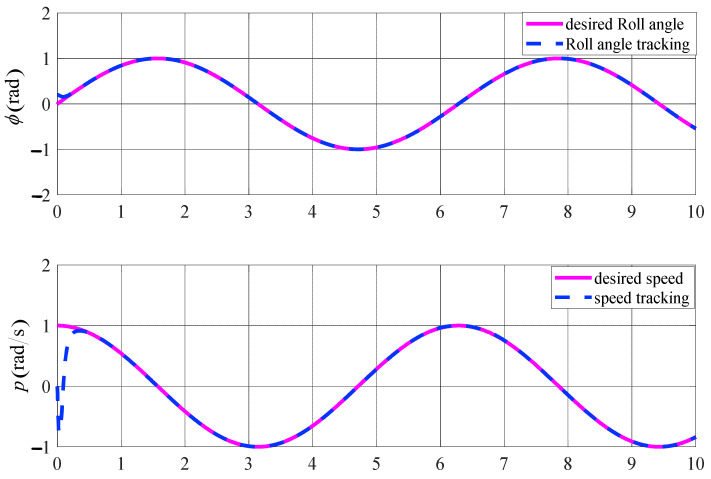
The performance of roll angle tracking.

**Figure 8 sensors-23-00574-f008:**
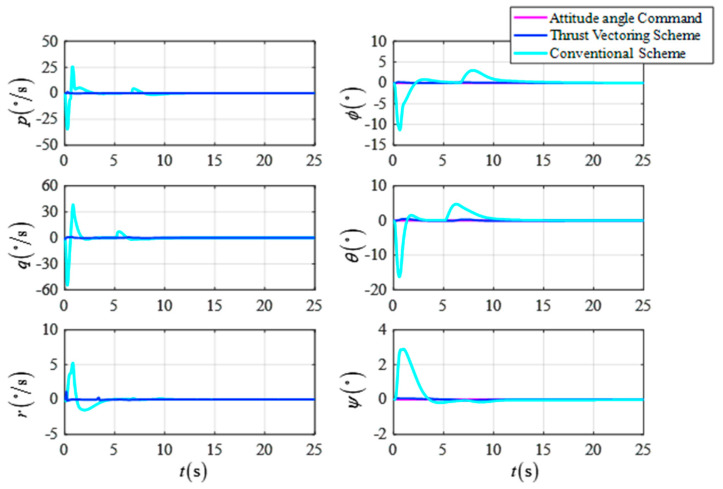
The attitude angle and the attitude angular velocity in the target point flight.

**Figure 9 sensors-23-00574-f009:**
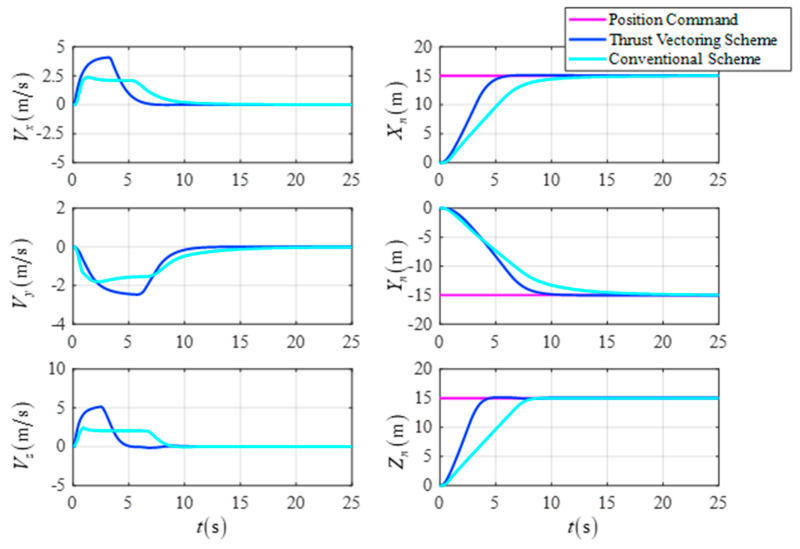
The position and the velocity in the target point flight.

**Figure 10 sensors-23-00574-f010:**
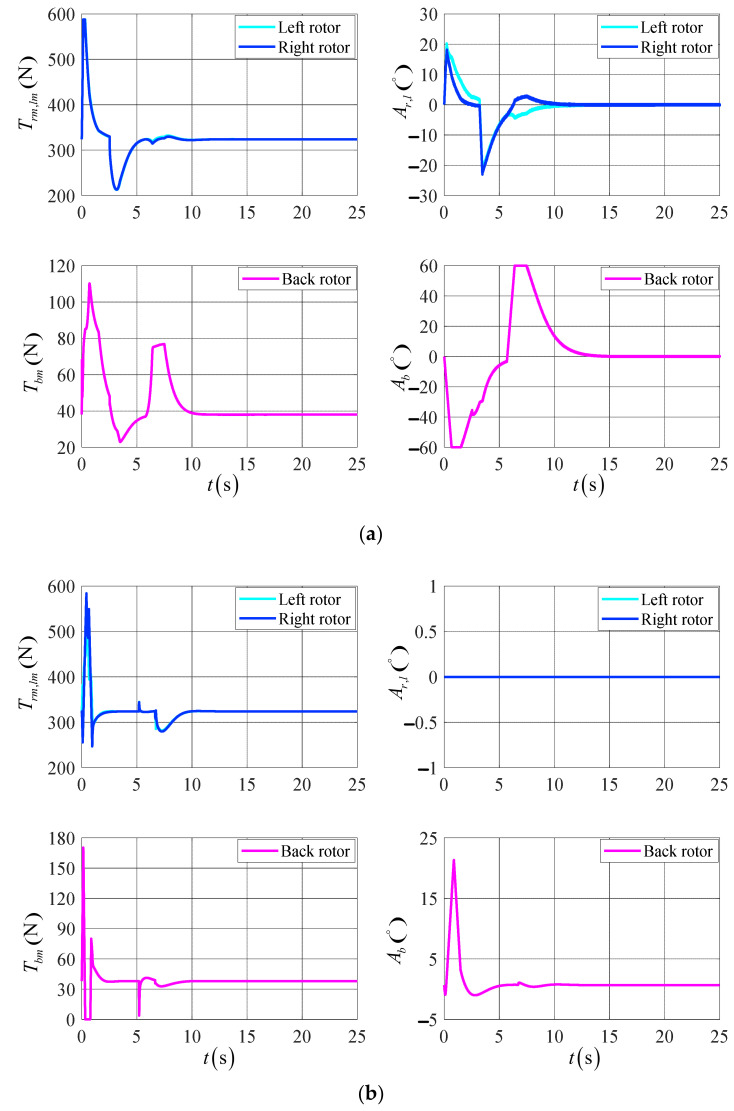
The variation of the UAV actuators’ control in the target point flight. (**a**) The results of the DTVC scheme; (**b**) the results of the conventional control scheme.

**Figure 11 sensors-23-00574-f011:**
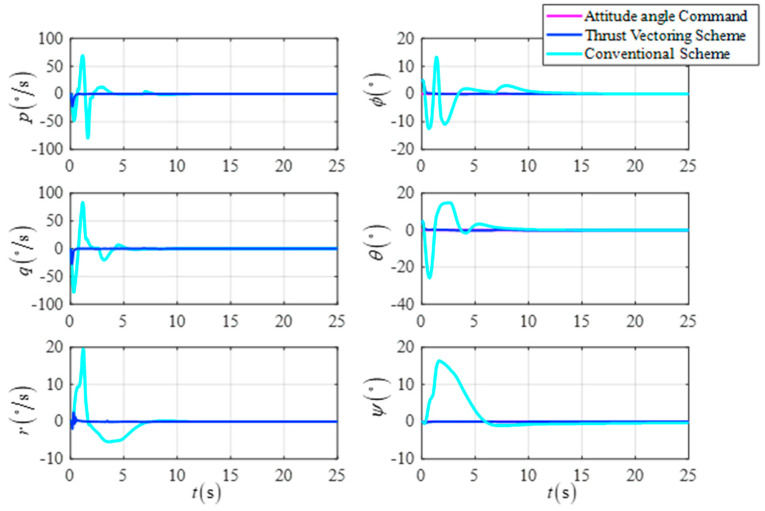
The attitude angle and the attitude angular velocity in the fixed height flight.

**Figure 12 sensors-23-00574-f012:**
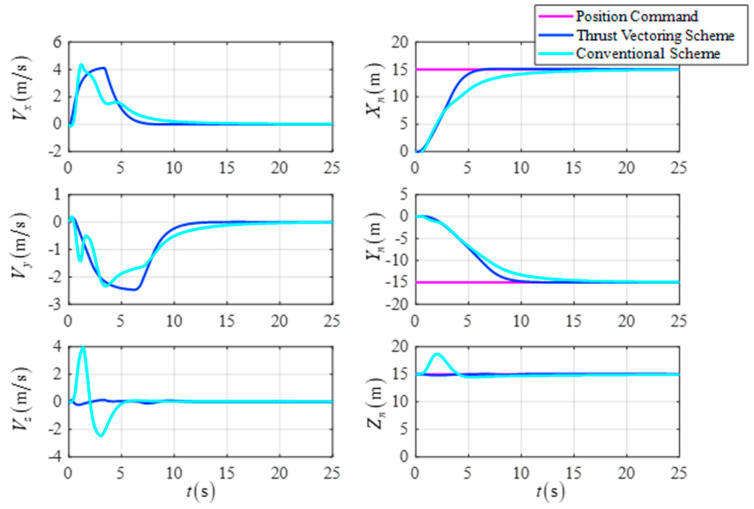
The position and the velocity in the fixed height flight.

**Figure 13 sensors-23-00574-f013:**
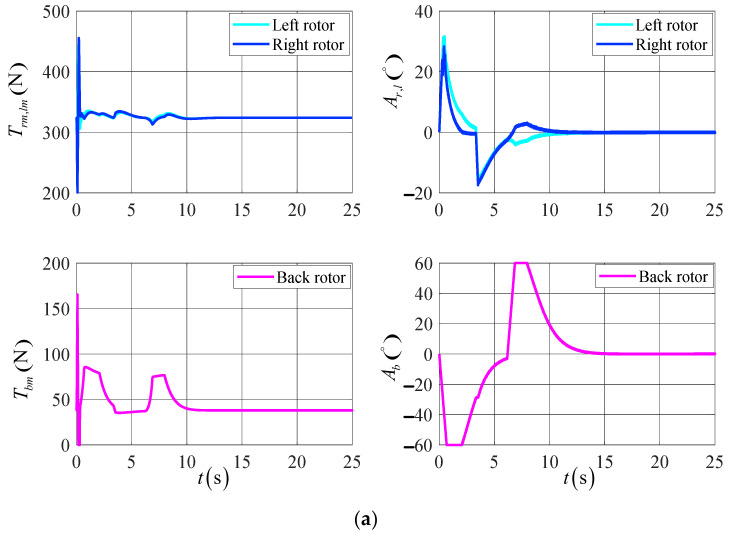
The variation of the UAV actuators’ control in the fixed height flight. (**a**) The results of the DTVC scheme; (**b**) the results of the conventional control scheme.

**Figure 14 sensors-23-00574-f014:**
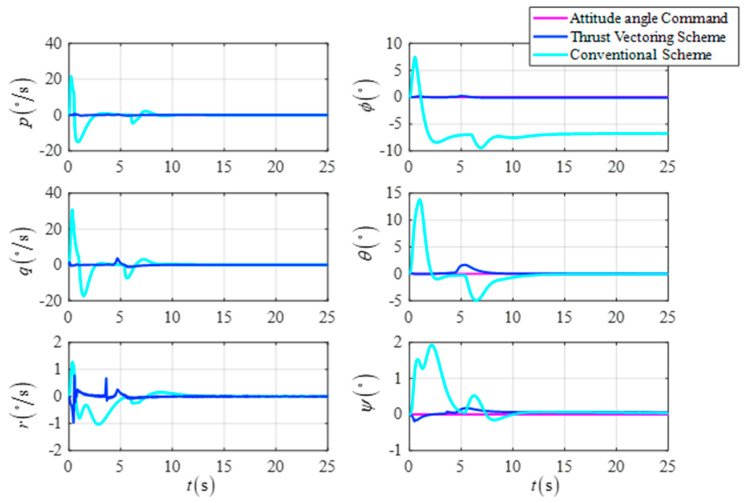
The attitude angle and the attitude angular velocity in landing.

**Figure 15 sensors-23-00574-f015:**
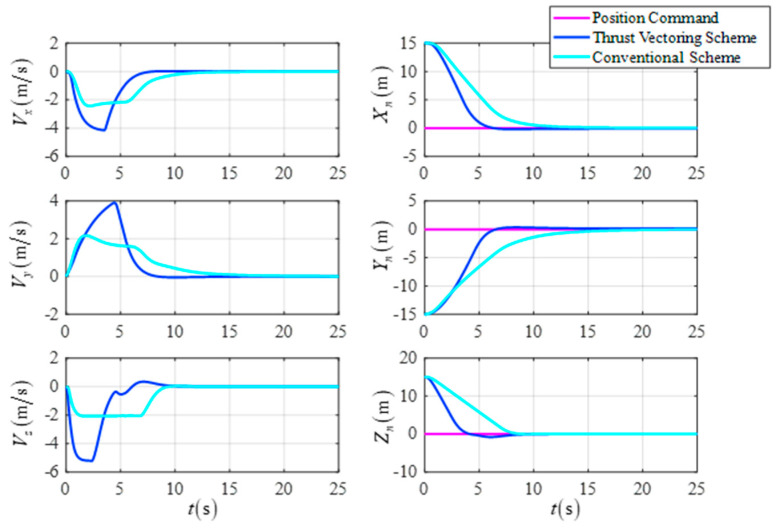
The position and the velocity in landing.

**Figure 16 sensors-23-00574-f016:**
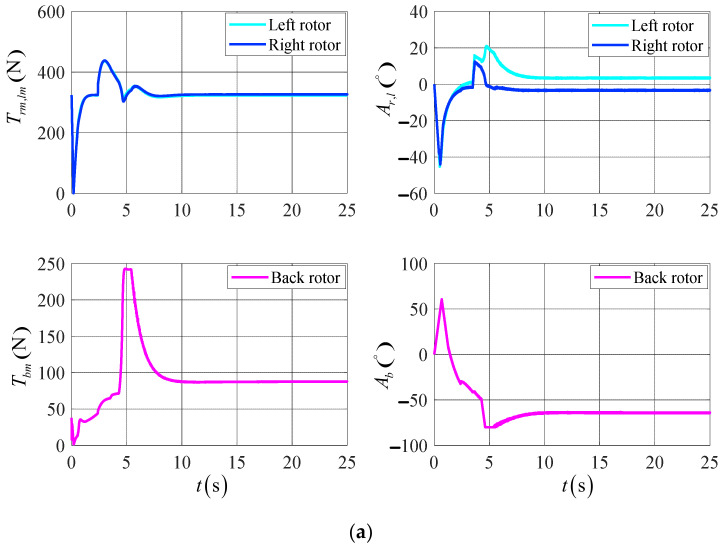
The variation of the UAV actuators’ control in landing. (**a**) The results of the DTVC scheme; (**b**) the results of the conventional control scheme.

**Figure 17 sensors-23-00574-f017:**
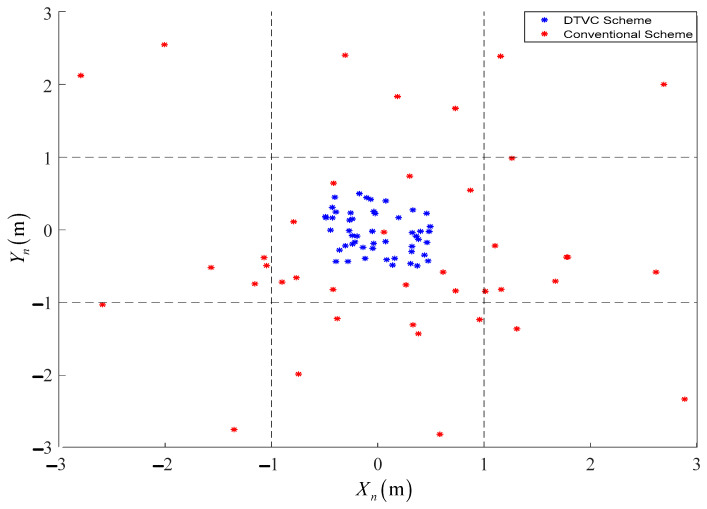
The statistical results of the landing locations for the two schemes.

## Data Availability

Not applicable.

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
