# Peer review of "Thrust Vectoring Control of a Novel Tilt-Rotor UAV Based on Backstepping Sliding Model Method"

_sensors, 2023, doi:10.3390/s23020574_

Round 1
Reviewer 1 Report
This paper presents a thrust vectoring control scheme for a tilt-rotor UAV based on backstepping sliding mode control method. Many grammatical mistakes are found which makes the paper difficult to read. The detailed comments are listed as follows:
1. The modeling of the propeller and ducted fan is missing.
2. It looks like this paper only investigates the control performance of the UAV under VTOL mode. Therefore, the title of the paper may be changed.
3. The control allocation equation as described in Eqs. (7), (9), (10) are kind of complicated. However, it is not clear that how the actual control inputs are calculated based on these equations. This part is the core of the paper, however, the present expression on this problem makes the paper hard to understand. The contributions can not be justified.
4. Definitions of some variables are missing.
5. From the simulation results, it looks like at the initial stage, the pitch moment of the UAV is not balanced.
6. The quality of the demonstrated figures is very low and hard to understand.
7. At balanced position, each of the front twol rotors provides more than 300N, and the rear rotor only provides 4N. Is this reasonable? Based on the demonstrated schematic of the UAV, this is not correct.
8. The performance of the compared control scheme is questionable.
Author Response
Dear Editors and Reviewers,
Thank you very much for your careful review and constructive suggestions with regard to our manuscript " Thrust Vectoring Control of a Novel Tilt-rotor UAV Based on Backstepping Sliding Model Method " (Manuscript ID: sensors-2046792). These comments are helpful for us to revise and improve our manuscript. We have studied comments carefully and tried our best to revise and improve the manuscript. Please see responses to the review comments in the accompanying file.

Reviewer 2 Report
1. The contributions of this paper is not clear. It seems just a change of control structure. The authors should summarize the contributions in Introduction.
2. Equation (23) is the virtual control law. Although the control allocation result is shown in Table 1, the description of control allocation is not enough. That is how to decouple multiple real control inputs.
3. In the control curves of simulations, the subtitles of each subfigures (a) and (b) should be given to tell their difference, e.g. Figure 10.
4. Please interpret crosswind disturbance by combining the UAV model.
Author Response

(The authors gave the same response as above.)

Reviewer 3 Report
Valuable contribution in the field of UAV's control, but in the future, for similar approaches it should be insisted on the construction and implementation of the control strategy.
Author Response

(The authors gave the same response as above.)

Round 2
Reviewer 1 Report
The authors have answered all my questions. I have no further comments.
Author Response
Dear Editors and Reviewers,
Thank you very much for your careful review and constructive suggestions with regard to our manuscript " Thrust Vectoring Control of a Novel Tilt-rotor UAV Based on Backstepping Sliding Model Method " (Manuscript ID: sensors-2046792). These comments are helpful for us to revise and improve our manuscript.
Reviewer 2 Report
The paper is revised according to the comments. It seems better than the previous. However, there some unclear descriptions, such as which are the real control variables: Tr, Tl, Tb and Trm, Tlm, Tbm?
Author Response

(The authors gave the same response as above.)
